# Endoscopic Management of Dysplastic Barrett’s Oesophagus and Early Oesophageal Adenocarcinoma

**DOI:** 10.3390/cancers15194776

**Published:** 2023-09-28

**Authors:** Leonardo Henry Eusebi, Andrea Telese, Chiara Castellana, Rengin Melis Engin, Benjamin Norton, Apostolis Papaefthymiou, Rocco Maurizio Zagari, Rehan Haidry

**Affiliations:** 1Gastroenterology Unit, IRCCS Azienda Ospedaliero-Universitaria di Bologna, 40138 Bologna, Italy; chiara.castellana@outlook.it (C.C.); renginmelis.engin@studio.unibo.it (R.M.E.); 2Department of Medical and Surgical Sciences, University of Bologna, 40138 Bologna, Italy; roccomaurizio.zagari@unibo.it; 3Digestive Disease and Surgery Institute Cleveland Clinic, London SW1X 7HY, UK; telesea@ccf.org (A.T.); benjamin.norton@nhs.net (B.N.); 4Division of Surgery and Interventional Science, University College London, London NW1 2BU, UK; 5Department of Gastroenterology, University College London Hospital (UCLH), London NW1 2BU, UK; a.papaefthymiou@nhs.net; 6Centre for Obesity Research, Department of Medicine, Rayne Institute, University College London, London NW1 2BU, UK; 7Esophagus and Stomach Organic Diseases Unit, IRCCS Azienda Ospedaliero-Universitaria di Bologna, 40138 Bologna, Italy

**Keywords:** Barrett’s oesophagus, endoscopic treatment, radiofrequency ablation

## Abstract

**Simple Summary:**

Among individuals with gastro-esophageal reflux disease, the prevalence of histologically confirmed BO is around 7%, with variations according to different geographical regions. Since Barrett’s oesophagus may progress to cancer through various stages of dysplasia, a correct diagnosis is pivotal in the management of patients with Barrett, made through accurate endoscopic examination and tissue sampling. The management of BO depends most strongly on the presence and severity of dysplasia, thus regular endoscopic surveillance and biopsies are required to monitor for neoplastic progression. In the presence of Barrett’s-associated neoplasia, endoscopic treatments are utilised, including resection techniques and ablation therapies, and long-term data support their safety and efficacy. However, they are not without risk, and for the optimal management of BO-associated neoplasia, it is recommended that patients are referred to expert centres.

**Abstract:**

Barrett’s oesophagus is a pathological condition whereby the normal oesophageal squamous mucosa is replaced by specialised, intestinal-type metaplasia, which is strongly linked to chronic gastro-oesophageal reflux. A correct endoscopic and histological diagnosis is pivotal in the management of Barrett’s oesophagus to identify patients who are at high risk of progression to neoplasia. The presence and grade of dysplasia and the characteristics of visible lesions within the mucosa of Barrett’s oesophagus are both important to guide the most appropriate endoscopic therapy. In this review, we provide an overview on the management of Barrett’s oesophagus, with a particular focus on recent advances in the diagnosis and recommendations for endoscopic therapy to reduce the risk of developing oesophageal adenocarcinoma.

## 1. Introduction

Barrett’s oesophagus (BO) is a pathological condition that occurs due to metaplastic change that starts within the distal oesophagus. Here, the normal squamous mucosa is replaced by specialised intestinal-type metaplasia (SIM) that contains goblet cells [1,2].

Barrett’s oesophagus is linked to gastro-oesophageal reflux disease (GORD) as an adaptive reaction to chronic reflux-induced damage [3]. The presence of reflux symptoms is significantly associated with an increased risk of BO, with the strongest association found between weekly reflux symptoms and long-segment BO [4]. Globally, about 15% of the general population experience reflux symptoms, with some areas reaching a prevalence of 50% [5,6]. However, among individuals with GORD, the prevalence of histologically confirmed BO is around 7%, with variations according to different geographical regions that range from 4% in Asian countries to 14% in North America [7,8].

Endoscopic screening may be considered in patients with chronic GORD symptoms who have at least three of the following risk factors: ≥50 years old, male sex, body mass index (BMI) ≥ 30 kg/m^2^, and Caucasian ethnicity [7]. Also, the presence of hiatal hernia should be considered a significant risk factor for BO, whereas no association has been found between *Helicobacter pylori* infection and either endoscopically diagnosed or histologically confirmed BO [9,10]. Nevertheless, if a family history of Barrett’s or oesophageal adenocarcinoma (OAC) is present, which should involve at least one first-degree relative, the threshold for screening is lower [11].

Barrett’s oesophagus may progress to OAC through various stages: from non-dysplastic BO (NDBO) to low-grade dysplasia (LGD), to high-grade dysplasia (HGD), and eventually to OAC. The estimated annual risk of progression to OAC in NDBO is 0.3%, in LGD it is 1%, and the highest risk is in HGD, when it increases to 8% [12,13].

A correct diagnosis is pivotal in the management of patients with BO that is made through accurate endoscopic examination and tissue sampling. This both helps to establish a formal diagnosis of BO and identifies subjects at high risk of progression towards cancer. The management of BO depends most strongly on the presence and severity of dysplasia, although other individual variables are important. Initial treatment relies on lifestyle modification to reduce acid reflux with medications that suppress stomach acid production. This is combined with regular endoscopic surveillance and biopsies to monitor for neoplastic progression. In the presence of Barrett’s-associated neoplasia, more invasive endoscopic treatments can be utilised, which include endoscopic mucosal resection (EMR) and endoscopic submucosal dissection (ESD) for the removal of visible dysplasia and endoscopic ablative techniques, including radiofrequency ablation (RFA) and cryotherapy, to eradicate non-visible dysplasia.

In this review, we discuss all aspects of the management of BO with a particular focus on recent advances in the diagnosis of SIM and the current evidence for endoscopic therapies to reduce the risk of progression towards OAC.

## 2. Diagnosis of Barrett’s Oesophagus

Under normal conditions, the oesophagus is lined with a stratified squamous epithelium, which has a light-coloured and glossy appearance upon endoscopic inspection [14]. The squamocolumnar junction, also known as the Z-line, is a macroscopically visible line that marks the contact between the squamous and columnar epithelium. This is noticeable in the distal oesophagus at the level of the gastro-oesophageal junction (GOJ). In BO, there is proximal displacement of the squamocolumnar junction away from the GOJ [14]. The metaplastic columnar mucosa of BO is easily visible by its reddish colour (often described as ‘salmon-coloured’) and velvet-like texture compared to the pale and glossy squamous mucosa [15].

The diagnosis of BO requires a combination of endoscopic and histologic criteria. This involves the recognition of the abnormal distal oesophageal lining at the time of upper gastrointestinal endoscopy, which is then supported by histological evidence of a columnar-lined epithelium and an oesophageal SIM [14,16]. Consequently, oesophagogastroduodenoscopy (OGD) is considered the current gold standard for the diagnosis of BO [17].

An accurate diagnosis of BO relies on the precise delineation of the GOJ during endoscopy. This allows for determining whether there is a proximal migration of the squamocolumnar junction leading to a columnar-lined section of the epithelium within the lower oesophagus. The most straightforward landmark to define the GOJ, and the recommended minimum requirement, is the proximal limit of the longitudinal stomach folds with minimal air insufflation [18,19]. Thus, the initial endoscopic diagnosis of BO can be carried out when the endoscopist detects the presence of ≥1 centimetre of salmon-coloured mucosa extending proximally beyond the GOJ on the top of the gastric folds [20]. The use of incorrect landmarks for the GOJ can lead to a misclassification of BO, with a negative impact on the early diagnosis of neoplasia [21].

Barrett’s oesophagus may appear during endoscopy as a lesion that extends segmentally or circumferentially. Once BO is identified, the Prague criteria are used to measure and classify its length, distinguishing between the circumferential extension (C) and the maximum longitudinal extension (M) of Barrett’s metaplasia [22]. When the columnar epithelium rises at least 1 cm above the GOJ, interobserver agreement between endoscopists using the Prague criteria is excellent. However, the interobserver agreement was found to be poorer for shorter BO segments [14]. Moreover, short-segment BO is defined as having ≤3 cm of metaplastic epithelium, while long-segment BO is defined as having >3 cm of metaplastic epithelium above the GOJ [19]. An extension of the columnar epithelium < 1 cm, and in the absence of any confluent columnar-lined segment, should be considered as an irregular squamocolumnar junction rather than BO [11]. Indeed, patients with GORD are more likely to harbour an irregular Z-line, and up to 40% of biopsies taken from an irregular Z-line may contain intestinal metaplasia, but the relevance of this finding is still to be established [23,24]. Since the diagnosis of an irregular Z-line is subjective, and there is no accepted length cut-off to distinguish between an irregular Z-line and BO, it is suggested that 1 cm (M of the Prague criteria) should be the minimum length for an endoscopic diagnosis of BO. In general, biopsies are not recommended when an irregular Z-line is encountered, but they may be performed to aid the diagnosis depending on the degree of suspicion. If the biopsy specimens are taken within an irregular Z-line with no clear endoscopic evidence of BO, they should then be labelled as GOJ and not oesophageal biopsy samples [11].

However, the accuracy of a standard endoscopic examination with biopsy sampling for BO diagnosis can be limited by several factors, including the endoscopist’s experience, the endoscopes definition, and the location of biopsies. Thus, to improve the diagnosis of oesophageal intestinal metaplasia and dysplasia, additional endoscopic procedures have been suggested. The ultimate objective of these newer methods is to enhance the endoscopic identification of curable Barrett’s-associated neoplasia while lowering the procedure time, cost, and sampling error [25]. One of these procedures is chromoendoscopy, which has been increasingly used to improve the yield of SIM in BO [26]. As the name suggests, it involves the use of dyes, such as methylene blue, Lugol’s iodine, indigo carmine, and acetic acid, which help to improve the detection rates by highlighting various features of the oesophageal mucosa [26,27]. Absorptive stains, such as Lugol’s solution and methylene blue, identify specific epithelial cell types by preferential absorption or diffusion across the cell membrane. Contrast stains, such as indigo carmine, seep through mucosal crevices and highlight surface topography and mucosal irregularities [28]. Lugol’s iodine is used for recognizing squamous tissue, squamous dysplasia, and squamous cell carcinomas because it preferentially stains the non-keratinised squamous epithelium [28]. This property makes it a good choice for staining oesophageal lesions as it is taken up by oesophageal squamous cells that contain glycogen [29]. However, it can be used to assess the success of the endoscopic therapy, as residual islands of Barrett’s metaplasia are not stained by Lugol’s iodine [29]. Methylene blue is a vital dye that may be used to detect BO because it is readily absorbed by columnar intestinal-type cells [30]. However, recent findings suggest that the detection of metaplasia by chromoendoscopy using methylene blue is not significantly different compared to the conventional four-quadrant biopsy technique, although the number of biopsies needed is significantly fewer [27,31]. Another factor to consider is the potential DNA-damaging effect of methylene blue on the Barrett’s epithelium that may discourage its use [32]. The concomitant use of carmine dye or acetic acid staining with magnification endoscopy may enhance the recognition of different mucosal pit patterns in the columnar epithelium [29]. Chromoendoscopy with vital staining has been demonstrated to identify more patients with short-segment BO. Short segments are associated with a low yield of intestinal metaplasia (30–50%) when biopsy specimens are randomly acquired [33]. While chromoendoscopy significantly increases the detection of intestinal metaplasia and limits the number of biopsies required in short-segment BO, it does not appear to be beneficial in patients with an irregular Z-line (i.e., <1 cm of columnar mucosa in the distal oesophagus) [33].

In recent years, virtual chromoendoscopy has become available, enabling more practical chromoendoscopy without the use of dyes. Several virtual chromoendoscopy technologies have been developed, including narrow-band imaging (NBI; Olympus, Tokyo, Japan), blue light imaging (BLI; Fujifilm, Tokyo, Japan), and i-Scan (PENTAX Medical, Montvale, NJ, United States), based on light filters or post-image acquisition processing [11]. In comparison to standard resolution endoscopy, virtual chromoendoscopy allows for the better visualisation of mucosal glandular and vascular structures. Most evidence has been accumulated on the NBI system (Figure 1), including a prospective tandem study, which demonstrated that NBI led to a significantly higher rate of both the detection and grade of dysplasia with fewer biopsies [34]. Moreover, a recent meta-analysis of six studies reported a high diagnostic accuracy of NBI with targeted biopsies for detecting dysplasia of all grades compared to standard white light endoscopy with a standard biopsy protocol in a per-patient analysis. The authors reported a pooled sensitivity of NBI of 76% (95%CI: 0.61–0.91) and a pooled specificity of 99% (95%CI: 0.99–1.00) [35]. However, the interobserver agreement for the interpretation of virtual chromoendoscopy imaging is not always optimal, which may be a limitation in clinical practice [36]. Nevertheless, the increasing evidence for the potential benefits of virtual chromoendoscopy for the screening and surveillance of BO has led to its use being recommended when inspecting Barrett’s segments [37].

Another recently developed diagnostic method is the confocal laser endomicroscopy (CLE) method. After an intravenous fluorescein injection, the oesophageal tissue is illuminated using a blue laser. This technique reproduces in vivo real-time imaging at a high magnification, allowing for the identification of suspicious lesions and for performing targeted biopsies [38]. A meta-analysis aiming to assess the accuracy of CLE for the diagnosis of neoplasia in BO, including more than 4000 lesions, showed a per-lesion pooled sensitivity and specificity of 77% (95%CI: 0.73–0.81) and 89% (95%CI: 0.87–0.90), respectively [39]. Thus, CLE appears to be very promising; however, its use is not currently recommended routinely but rather as an adjunctive imaging technique to identify dysplasia and cancer in select BO cases in expert centres [37].

When a diagnosis of BO is suspected during endoscopy, the endoscopist should perform biopsies following the “Seattle protocol”. The presence of dysplasia within Barrett’s mucosa is often patchy [40], which causes oesophageal biopsies to have a significant sampling error [41]. This protocol was designed to minimise the chance of missing a concealed lesion, which may be randomly distributed along the Barrett’s epithelium. This protocol requires taking four-quadrant biopsy samples at every 1–2 cm intervals throughout the columnar-lined oesophagus. In addition, areas of any mucosal irregularity such as masses, nodules, and ulcerations must be sampled, as they are associated with a greater likelihood of harbouring dysplastic tissue [25]. The adherence to recommended surveillance procedures, such as the Seattle protocol, is associated with a higher dysplasia detection; however, it requires a lot of time, effort, and money and is still prone to sampling errors [41]. Therefore, it is unsurprising that adherence to such a protocol was found to be low among endoscopists, and adherence was inversely related to the length of the BO segment [41].

In Europe, the most widely used grading system for the histopathological diagnosis of BO-associated dysplasia is the revised Vienna classification. This original system was developed to standardise the terminology for the histological grading of gastrointestinal mucosal neoplasms. This was because of discrepancies in the grading systems used around the world for the categorisation of early neoplastic lesions. The system is versatile, dividing early mucosal lesions into one of five categories, and can be used for other gastrointestinal epithelial neoplastic or dysplastic lesions [42,43].

Three different types of columnar epithelia can be found in BO: a cardia-type epithelium almost completely composed of mucus-secreting cells; a gastric fundic-type epithelium with mucus-secreting cells, parietal cells, and chief cells; and an intestinal-type epithelium that is characterised by the presence goblet cells [44]. BO fundic- and cardia-type epithelia might morphologically look identical to the stomach’s columnar epithelia. However, most of the professional guidelines concur that SIM is a necessary element for a formal diagnosis of BO [45].

Upon histological analysis, cells with BO-associated LGD exhibit mild architectural defects, an increased number of mitoses, and cytologic atypia, which includes an elevated nuclear/cytoplasmic ratio and nuclear elongation [46]. In the presence of LGD on random biopsies, the diagnosis should be confirmed by a second expert GI pathologist and referred to an expert centre [11,47,48,49,50]. It is also recommended to confirm the diagnosis of LGD after a surveillance interval of 6 months before offering endoscopic treatment [47,48,49]. Similarly, the diagnosis of HGD should be confirmed by a second expert GI pathologist and referred to a BO expert centre. Here, a repeat high-definition endoscopy should be completed to identify and treat all visible abnormalities in the dysplastic mucosa [48]. Therefore, the timing of endoscopic treatment is governed by the grade of dysplasia, and the visible characteristics of the dysplastic mucosa will guide to the most appropriate endoscopic therapy in the form of ablation or resection (Table 1).

## 3. Endoscopic Resection Techniques for Dysplastic Barrett’s and Early Oesophageal Adenocarcinoma

The main goal of an accurate endoscopic examination is to identify any suspicious lesions that require removal. For visible neoplasia, endoscopic resection (ER) is recommended. Elevated lesions are more likely to harbour neoplasia in comparison to flat lesions, but both require ER before ablation to increase the success of remission [52]. Moreover, one of the advantages of ER over ablation is that it allows for an accurate histological examination of the whole lesion. Indeed, changes in the histological stage after ER have been reported in up to a third of cases compared to initial biopsies, thus ensuring optimal management [17].

The two main ER techniques are EMR and ESD. The recent European Society of Gastrointestinal Endoscopy (ESGE) guidelines suggest using EMR for lesions ≤20 mm that have a low probability of submucosal invasion (i.e., Paris type 0-IIa, 0-IIb) and for larger or multifocal benign lesions. ESD should be performed for lesions suspicious for submucosal invasion (i.e., Paris type 0-Is, 0-IIc), for malignant lesions >20 mm, and for lesions in scarred or fibrotic areas [53]. Moreover, for deep excavated lesions (Paris type 0-III), ER is not recommended due to the high risk of deep invasion and lymph-node metastasis [54].

Early mucosal OAC, known as T1a, has a low risk of lymph node metastasis on systematic review (estimated around 2%), which means early, small lesions may be eligible for EMR [55]. A trial on 107 patients with BO-related lesions who were eligible for endoscopic treatment showed complete endoscopic eradication with EMR in 80% that included a T1a OAC in 36% [56].

With EMR, smaller lesions are usually resected en-bloc, whereas larger ones require multiple resections using the so-called piecemeal approach. Endoscopic mucosal resection can be performed using two main techniques: cap-snare and band-ligation. With the cap-snare technique, the lesion is first lifted and then drawn into the cap and resected by a snare. The band-ligation technique involves the release of an elastic band at the bottom of the lesion, which generates a pseudopolyp that can then be resected by a hot snare (Figure 2).

The efficacy of both EMR techniques is comparable, although band-ligation is most commonly performed given its comparable ease and shorter time [20]. A comparative study between the two techniques has shown no significant differences in the maximum diameter of the resected specimen (approximately 16 × 11 mm for band-ligation versus 15 × 10 mm for cap-snare) nor any differences in the maximum diameter of the resected ulcer base after 24 h (approximately 21 × 14 mm for band-ligation versus 19 × 13 mm for cap-snare). In the same study, the overall complication rate was 2%, and the failure rate of ER was 7% with no significant differences between the two groups [57].

The ESD technique implies the use of an electrosurgical knife to dissect the submucosa underneath the lesion. Submucosal dissection is achieved after the injection of a lifting agent, which subsequently enables an en-bloc resection. This technique is particularly useful for the resection of large oesophageal lesions. Indeed, a higher percentage of en-bloc resections and subsequent lower rates of recurrence have been reported with ESD over traditional EMR [20,54].

Compared to surgery, ER appears effective for patients with dysplastic BO and early T1a OAC with a better safety profile [20]. In a meta-analysis of 7 studies that included 870 patients with HGD or T1a OAC, there was no significant difference in neoplastic remission or overall survival between surgery and endotherapy (ER and ablation). There was a higher rate of major adverse events in those undergoing surgery, whereas those having endotherapy had a higher rate of neoplasia recurrence, although most could be retreated successfully with endotherapy [58].

Endoscopic resection with ESD may be an alternative to surgery for T1b OAC, especially in patients who are poor surgical candidates. This is particularly true when the risk of lymph node metastasis is deemed low. Histopathological characteristics of T1b tumours associated with a low risk of lymph node metastasis include a tumour infiltration depth < 500 µm, the absence of poor differentiation, the absence of lymphovascular invasion, and clear deep resection margins (R0) [48]. The ability to achieve an R0 resection with ESD was determined in a retrospective cohort study that showed a rate of 87% for T1a OAC and 49% for T1b OAC [59].

A randomised clinical trial compared EMR to ESD for 40 patients with BO and HGD or early OAC, reporting that both ER techniques appeared to be highly effective in terms of the need for surgery, neoplasia remission, and recurrence [60]. Moreover, ESD achieved significantly higher en-bloc (R0) resection rates compared to EMR (59% vs. 12%), but the overall remission rates at 3 months were similar (94% vs. 94%); ESD was, however, more time consuming and caused severe AE more frequently [60].

A recent prospective study on 537 patients who underwent cap-assisted EMR or ESD followed by ablation showed that complete remission of dysplasia was higher in the patients treated with ESD compared to EMR at 2 years. Complete remission of intestinal metaplasia was similar in both groups, and there were no significant differences in complications [61].

The most common complication of ER is an oesophageal stricture, followed by bleeding and perforation. Strictures are related to the extension of the resected mucosa, with the risk increasing as the length of the circumferential resection increases. Perforation rates after EMR and ESD range between 0–5% and appear to be higher with ESD. Bleeding is common, but it is usually controlled with endoscopic haemostatic treatment. Admission for uncontrolled bleeding after ER is rare [62]. Konda et al., reported on the rates of complications among patients undergoing EMR. The rate of stricture formation was 41.5% (38% symptomatic), bleeding 3%, and perforation 19% [56].

After successful resection, recurrence rates up to 30% over 3 years have been observed for patients in whom the remaining BO is left untreated. Moreover, several studies have shown benefits in the outcomes of patients treated with RFA compared to endoscopic surveillance without any treatment, resulting in a reduced risk of neoplastic progression after RFA [63,64]. Therefore, endoscopic ablation is currently a guideline recommendation to achieve the complete eradication of all the remaining BO after ER of visible lesions [47,48,49,50,51].

Overall, straightforward cases should be managed following the current guidelines for BO follow-up and treatment (Figure 3). However, more complex cases require a case-by-case evaluation, taking into account several factors including institutional experience, the lesion’s characteristics, and the patient’s comorbidities and preferences, and they should therefore be discussed at MDT meetings in order to establish the most appropriate management.

## 4. Ablation Treatments for Barrett’s Oesophagus

Endoscopic ablation of BO aims to destroy the abnormal mucosa to prevent further neoplastic progression. When discussing ablation techniques, it is crucial to emphasise that tissue disruption is limited to the mucosa. In addition, tissue coagulation prevents the acquisition of tissue for histological characterisation. This means that the endoscopist must be confident that the disease is limited to the mucosa to ensure complete eradication and avoid luminal or extraluminal recurrence.

Historically, several techniques have been used with different degrees of success. In the early 1990′s, cases of BO ablation were reported using a neodymium-doped yttrium aluminium garnet (YAG) argon laser and photodynamic therapy [65,66,67,68,69]. These techniques were progressively abandoned in favour of argon plasma coagulation (APC) and multipolar electrocoagulation that were introduced later in the decade [70,71]. Their role in clinical practice remained controversial and not defined for almost a decade; in 2004, the American Gastroenterological Association workshop recognised the potential role of mucosal ablation in a subgroup of Barrett’s patients. However, the selection criteria for patients that might benefit from mucosal ablation were not discussed by the working group [72]. More robust data were collected with the introduction of radiofrequency ablation (RFA), and the role of mucosal ablation was progressively recognised and advocated for by societal guidelines from the late 2000s and early 2010s [11,73,74].

At present, endoscopic ablation of BO is recommended for the treatment of residual BO following resection of any visible lesions and in patients with confirmed LGD to reduce the risk of progression toward more advanced neoplastic alterations, such as HGD and OAC [11,47,50].

As mentioned, several ablation techniques have been implemented over the last decades that can be carried out either by heating (e.g., RFA or APC) or freezing (e.g., cryoablation) to destroy the Barrett’s epithelium. RFA is the most widely used in clinical practice; this technique uses a bipolar electrode in direct contact with the oesophageal mucosa to generate heat and induce a coagulative necrosis of the targeted mucosa. Among all the ablation techniques, RFA has the largest body of evidence, and its safety and efficacy has been evaluated in several studies including randomised trials and meta-analyses [75].

In 2009, Shaheen et al., published a randomised trial comparing ablation in BO-associated dysplasia compared to a control group using RFA. The rate of eradication in the patients with LGD was 90.5% compared with 23% in the control group, and the rate of eradication of HGD was 81% compared with 19% in the control group. The patients treated with RFA also had a significantly reduced disease progression (4% vs. 16%) and fewer cancers (1% vs. 9%), albeit with a stricture rate of 6% [76].

A subsequent large cohort study was published reporting the outcomes of 335 patients from the UK National Halo RFA Registry. This demonstrated eradication rates of 81% among all cases of BO-associated dysplasia after 12 months of treatment with a better response for short-segment BO [77]. The same authors showed an improvement in dysplasia and intestinal metaplasia clearance rates over the 6 years of observation from 77% to 92% and from 56% to 83%, respectively (*p* < 0.0001). In addition, the study demonstrated an increase in ER using EMR for visible lesions from 48% to 60% (*p* = 0.013) and a reduction in the rescue EMR following RFA from 13% to 2% (*p* < 0.0001). However, progression to OAC at 12 months remained statistically non-significant (3.6% vs. 2.1%, *p* = 0.51) [78].

Similarly, in 2014, Phoa et al., published the results of the SURF study showing that RFA is effective in treating LGD and eradicating BO. They achieved a complete eradication of dysplasia in 93% vs. 28% in the control group and a complete eradication of intestinal metaplasia in 88% vs. 0% in the control group (*p* < 0.001). This resulted in a reduced risk of progression to HGD or OAC by 25% and 7%, respectively. This study also showed that RFA presents an acceptable safety profile, with the most common adverse event being stricture formation that occurred in 12% of cases, which were treated with endoscopic dilatation [63]. The same cohort was analysed retrospectively after a median follow-up of 73 months in the study by Pouw et al. They reported a reduction in the absolute risk of BO progression following RFA of 32%, with only one case of progression to HGD/OAC in the RFA group (1.5%) during the follow-up compared to 23 cases in the surveillance group (34%). RFA achieved a complete BO clearance in 75 out of 83 patients, giving an eradication rate of 90%. Following RFA eradication, BO recurred in seven (9%) patients, of which three (4%) were diagnosed with LGD [64].

More recently, a prospective randomised study by Barret et al., showed a modest reduction in LGD and risk of progression at 3 years; indeed, the prevalence of LGD was 34% in the RFA group vs. 58% in the surveillance group (OR = 0.38; *p* = 0.05). Neoplastic progression was significantly higher in the surveillance group at 26% versus 12.5% in the RFA group (*p* = 0.15). A total of 22 adverse events were reported in the RFA group including bleeding or oesophageal stricture formation compared to no adverse events in the surveillance group. For this reason, the authors concluded that there was reduction in LGD prevalence and progression risk at three years The modest results in their study suggested that the risks and benefits of ablation should be weighed up carefully before proceeding to treat LGD dysplasia given the not-insignificant risk of complications after RFA [79].

In order to summarise these results, a recent meta-analysis by Shaheen [76], Phoa [63], and Barret [79] concluded that the pooled rate of progression of LGD to HGD or OAC was significantly lower in the RFA group than with endoscopic surveillance (RR 0.25; *p* = 0.04); however, the pooled risk of progression of LGD to OAC was slightly lower but not statistically significant (RR 0.56; *p* = 0.65). The patients in the RFA group also presented higher rates of complications including fever, bleeding, vomiting, nausea, and oesophageal strictures; hence, treatment options should be carefully weighed given the potential risk of oesophageal strictures following RFA treatment [80].

These data would suggest that, given the risk of potential complications and the heterogeneous rate of eradication, patient and centre selection remains paramount. In a retrospective analysis conducted in a French high-volume centre, including 96 consecutive patients with BO treated for dysplasia, there was a 59% rate of complete intestinal metaplasia eradication, a 79% rate of complete eradication of dysplasia, and a structure rate of 14% following RFA [81].

Catheter selection is another important aspect that is important to avoid overtreatment. There are different RFA catheters that can be used in practice depending on the clinical needs. In most scenarios, an over-the-scope RFA catheter can be used, which may include the Barrx 60, 90, and ultra-long catheters (Figure 4). These catheters differ in the dimension of the bipolar electrode, which includes a 15 mm long by 10 mm wide (Barrx 60), a 20 mm long by 13 mm wide (Barrx 90), and a 40 mm long by 13 mm wide (Barrx ultra long) electrode. A through-the-scope device is also available (Barrx channel 15.7 mm long by 7.5 mm wide) that has a flexible bipolar electrode, which is folded and passed into the working channel of a standard gastroscope. This device is particularly useful to target smaller Barrett’s segments or treat patients with strictures that might not accommodate larger over-the-scope devices. For long circumferential segments, the use of a Barrx 360 express device can be considered. This lies separate to the scope and is inserted over a guidewire. It consists of a self-inflating and self-sizing balloon that is covered by a winding RFA electrode measuring 4 cm in length.

Among these, the Barrx 360 express might present the highest rate of patient-related and device-related adverse events. In a recent post-marketing surveillance data analysis from August 2011 to August 2021 from the Food and Drug Administration’s Manufacturer and User Facility Device, a total of 87 patient-related adverse events including 15 strictures (17.2%), 13 mucosal laceration (14.9%), and 10 episodes of chest pain (11.4%) were reported in addition to 78 device-related malfunctions for RFA devices. The Barrx 360 express was involved in 61% of patient-related adverse events and 67% of device malfunction events; all the 15 oesophageal strictures secondary to treatment occurred with circumferential ablation devices [82].

Argon plasma coagulation is one of the first methods that was used to treat BO. Technically, APC is a non-contact thermal ablation technique that uses a through-the-scope catheter to deliver argon gas to the targeted mucosa. The gas is ionised when in contact with a high-voltage current on the tip of the catheter, and the resulting plasma causes thermal tissue coagulation. In 2006, the APBANEX prospective multicentre study showed a 77% rate of complete eradication of non-neoplastic BO treated with APC (90 W) in combination with esomeprazole 80 mg/day [83]. Following this, a randomised pilot study (the BRIDE study) has suggested that APC might have a similar efficacy and safety compared to RFA for the treatment of BO with HGD or OAC, with a more favourable cost difference [84].

APC can also be delivered following a submucosal injection of saline; this technique is known as hybrid APC (H-APC). The proposed advantage of the submucosal injection is to insulate and protect the subepithelial layers of the oesophagus, resulting in a lower stricture rate. An ex-vivo animal study showed that H-APC could reduce the coagulation depth compared to traditional APC and minimise thermal injury to the submucosal and muscular layers [85]. The same authors conduced a pilot study showing that H-APC achieved a complete macroscopical remission in 48 out of 50 treated patients (96%) after a median of 3.5 sessions (range 1–10). In this study, the histopathological eradication of BO was 78%, and the stricture rate was 2% [86].

Additional pilot studies and case series are available on the use of H-APC. The largest prospective study to date enrolled 146 patients and reported that, after 2 years, a total of 85 (66%) patients presented no recurrence of BO. In this study, H-APC showed an adverse event rate of 6%, with a stricture rate of 4% [87].

A recent systematic review and meta-analysis, pooling data from seven studies on H-APC, showed an overall complete remission rate of intestinal metaplasia of 91%, with an overall adverse event rate of 3%, including a stricture rate of 2%. However, as mentioned by the authors, the inclusion of non-controlled studies, retrospective cohorts, and case series might have lowered the overall quality of evidence [88]. Therefore, APC appears to be a safe and effective technique for the treatment of dysplastic Barrett’s; however, its use might be limited when long segments of BO need to be eradicated. Therefore, this technique should be considered in cases of short or focal segments of metaplasia or those refractory to RFA [89].

Other methods based on cold ablation have also been tested. The most recent literature on cold ablation refers to a C2 cryoballoon ablation system (CbAS). This consists of a though-the-scope self-inflating, self-sizing balloon catheter, which is attached to a controller that regulates a flow of nitrous oxide into the balloon to freeze the oesophageal mucosa. Cellular disruption is obtained following intracellular ice crystal formation that alters the cellular architecture. This technique does not require energy generation and might be particularly suited to treat segments of BO in patients that have strictures which cannot be traversed with an RFA catheter. Initial evidence showed that CbAS is safe and effective for the treatment of short-segment BO, with a 95% rate of complete eradication of dysplasia and intestinal metaplasia [90]. In a larger study including a total of 120 patients with LGD, HGD, and intramucosal adenocarcinoma, CbAS achieved a complete eradication rate of dysplasia and intestinal metaplasia of 97% and 91%, respectively, with a stricture rate of 12.5% [91]. A systematic review and meta-analysis of 272 patients showed a pooled rate of compete eradication of intestinal metaplasia and dysplasia of 86% and 94%, respectively, with an adverse event rate of 12.5% [92]. A retrospective study showed a comparable outcome for dysplastic BO treatment compared to RFA, with a possible higher stricture rate (10.4% vs. 4.4% *p* = 0.04) [93]. A large, prospective, European, multicentre study (EURO-COLDPLAY) is investigating the efficacy and safety of a focal cryoballoon for the treatment of BO, and an interim analysis has suggested using an 8 s rather than a standard 10 s duration of treatment. This is because the BO regression rates were similar, but there was a theoretical lower rate of stricture formation [94]. Novel cryoballoon devices should be implemented soon to treat larger segments of BO [95,96].

## 5. Failure of Therapy

Despite several advancements, the treatment of BO with a single technique might not achieve complete eradication. Therefore, a multimodal approach might be required for selected patients. Mittal et al., suggest considering further diagnostic tests and a multimodal approach in patients without a significant response after three sessions [97]. van Munster et al., retrospectively analysed the outcomes of patients undergoing RFA from the nationwide Dutch registry. In their study, 134 out of 1386 patients had poor mucosal healing following RFA that could be resolved with appropriate acid suppression and additional time; indeed, 67 of those 134 patients (50%) had normal squamous regeneration, achieving a complete eradication of BO in 97% of cases. These rates are similar to patients presenting with normal mucosal healing, whereas the remaining 67 patients (50%) had poor healing followed by poor squamous regeneration. A total of 74 out of 1386 patients (5%) with poor squamous epithelial regeneration had a higher risk of treatment failure (64% vs. 2%) and an increased risk of disease progression (15% vs. <1%) when compared to patients achieving normal mucosal regeneration. This study also identified risk factors independently associated with poor squamous regeneration such as higher body mass index, longer BO segments, reflux oesophagitis, and <50% squamous regeneration after baseline ER [98]. The same authors developed a model that identified other poor prognostic indicators associated with complex treatment courses such as those with a BO length ≥ 9 cm, the presence of HGD or OAC, and poor squamous epithelial regeneration [99].

## 6. Follow-up after Ablation

Long-term data from the UK National RFA registry show a risk of cancer at 10 years after ablation of 4% and a recurrence of dysplasia and intestinal metaplasia at 8 years of 6% and 19%, respectively. Nevertheless, most cases were treatable with the same modality [100].

Several post-RFA endoscopic surveillance intervals have been proposed; at present, the ACG guidelines are felt to be the most cost-effective strategy, suggesting endoscopic surveillance at 6 and 12 months followed by annual surveillance for patients with LGD, and a more intense surveillance at 3, 6, 9, 12, 18, and 24 months followed by annual surveillance for patients presenting with HGD [101].

van Munster et al., developed a prediction model of dysplasia recurrence after analysing data from 1154 patients during a mean follow-up of 4 years. During this time, a total of 38 patients developed recurrent disease (1% per person-year), and the authors identified some factors associated with recurrence such as the presence of new incident visible lesions during the treatment phase, a high number of endoscopic mucosal resections, male sex, an increased length of BO, the presence of HGD or OAC at baseline, and younger age [102]. This study could pave the road to further studies aiming to define predictors of recurrence for a more personalised surveillance strategy. This is particularly important because patients with BO-associated neoplasia are often frail and co-morbid and, following a successful endoscopic eradication therapy, are more likely to die from non-OAC causes [103].

## 7. Medical Management after the Endoscopic Treatment

Effective acid-suppression is considered to be an important condition for mucosal healing and squamous regeneration of BO following endoscopic therapy. The AGA expert review recommends the use of a proton pump inhibitor twice daily, and this was also further highlighted in a more recent paper by van Munster et al. [49,98]. H2-receptor agonist and sucralfate have also been used in European centres [104]; however, comparative studies on these drug regimens are lacking, and no definitive recommendation has been given.

Pain relief is another aspect to take into consideration following endoscopic intervention. It can usually be achieved with painkillers such as paracetamol, but there is no formal societal guidance on this issue. Similarly, no recommendations are available on diet; however, maintaining a liquid/soft diet in the days following the intervention might be reasonable to minimise the chances of traumatic injury.

Medical prevention of oesophageal strictures following the endoscopic treatment of BO is another aspect that has been investigated. A network meta-analysis conducted in 2019 showed that oral steroids might prevent postoperative strictures [105]. A more recent study evaluating the role of topical budesonide in patients undergoing oesophageal EMR or ESD showed conflicting results: no significant difference in stricture rates was seen in the patients taking topical budesonide compared to the patients not taking steroids (16% vs. 28%; *p* = 0.23); however, a logistic regression analysis taking into account potential confounders showed that the stricture rate was significantly lower (91%; 95%CI 0.0084–0.573; *p* = 0.023) in the budesonide cohort. The authors therefore suggest caution against concluding that budesonide is not effective, highlighting that, in their multivariate analysis, budesonide was associated with a lower stricture rate and concluding that budesonide might have a role in preventing stricture formation following oesophageal EMR and ESD [106].

Finally, with regards to the long-term management of patients with BO, other anti-reflux measures, such as surgical and endoscopic procedures, could be considered to address chronic gastroesophageal reflux insult, particularly in young patients who would require life-long endoscopic follow-up or for subjects who do not tolerate PPIs. However, anti-reflux surgery is not currently recommended by the ACG guidelines as an antineoplastic measure in patients with BO [50].

## 8. Conclusions and Future Directions

In the Western world, the incidence of BO is increasing alongside the rise in gastroesophageal reflux disease, its major risk factor. Although only a minority of patients with BO will progress to oesophageal adenocarcinoma, identifying high-risk individuals is pivotal, because oesophageal cancer is still associated with a high five-year mortality and associated care costs for healthcare systems. Future efforts should focus on improving endoscopists’ adherence to guidelines, with particular attention to the diagnosis of BO that includes the use of advanced endoscopic imaging and appropriate surveillance intervals.

Endoscopic endotherapy using both resection and ablation techniques have long-term data supporting their safety and efficacy; however, they are not without risk, and for the optimal management of BO-associated neoplasia, it is recommended that patients are referred to expert centres.

## Figures and Tables

**Figure 1 cancers-15-04776-f001:**
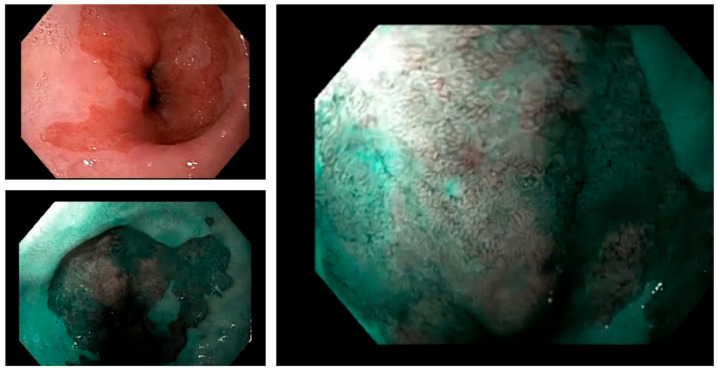
White light endoscopy, virtual chromoendoscopy (NBI), and virtual chromoendoscopy (NBI) with magnification images of Barrett’s oesophagus.

**Figure 2 cancers-15-04776-f002:**
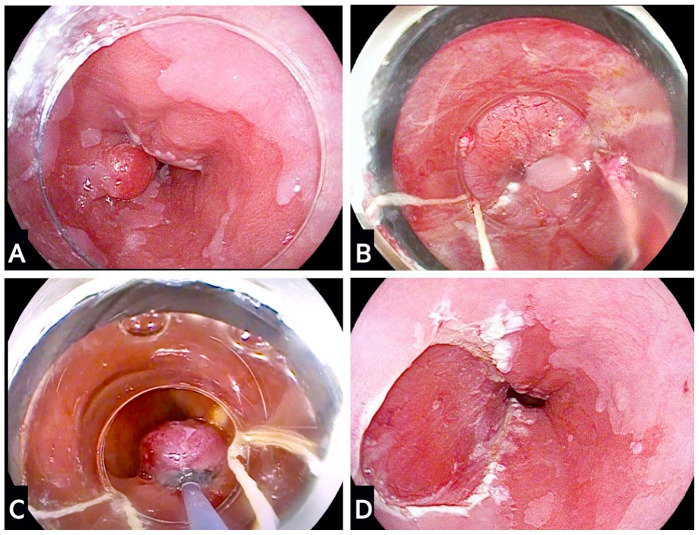
Endoscopic mucosal resection using multiband mucosectomy technique. (**A**) Evidence of a Paris Is nodule at 9 o’clock within a segment of Barrett’s oesophagus; (**B**) plan for endoscopic mucosal resection using a multiband mucosectomy set that is applied over-the-scope; (**C**) the nodule is banded and a snare is placed around the lesion to enable resection; (**D**) final endoscopic view of the resection base.

**Figure 3 cancers-15-04776-f003:**
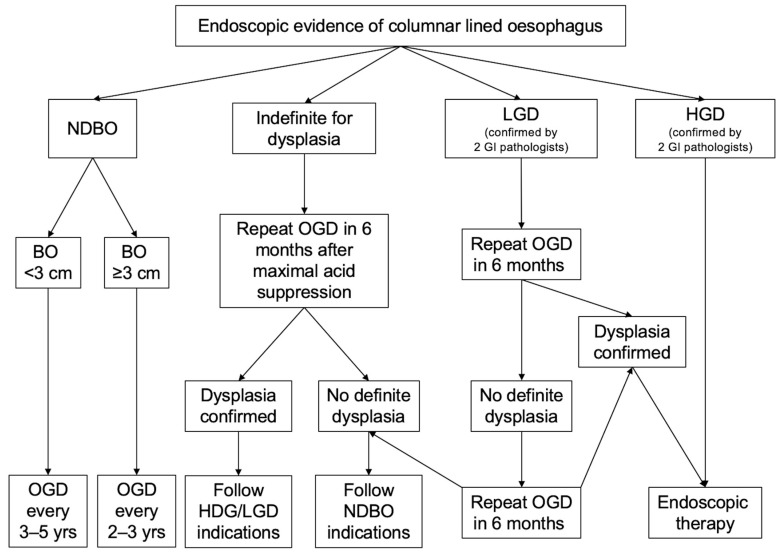
Endoscopic management of Barrett oesophagus based on the BSG guidelines [47].

**Figure 4 cancers-15-04776-f004:**
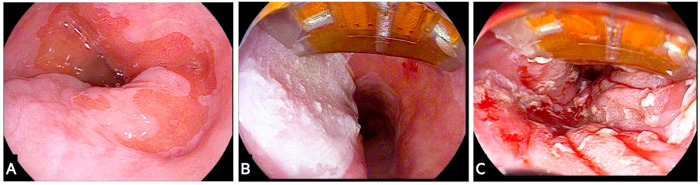
Radiofrequency ablation of dysplastic Barrett’s segment using a focal catheter. (**A**) Evidence of dysplastic Barrett’s segment undergoing ablation therapy; (**B**) over-the-scope ultra Barrx 90 focal RFA catheter with an area of ablated mucosa at 9 o’clock; (**C**) final endoscopic view following ablation of all Barrett’s segments.

**Table 1 cancers-15-04776-t001:** Guidelines on Barrett’s oesophagus management.

	Scientific Society	Authors, Year [Ref]	ND-BO	LGD-BO	HGD-BO
European	BSG	Fitzgerald et al., 2014 [11] Di Pietro et al., 2018 [47]	If maximum length < 3 cm, repeat OGD every 3–5 years. If maximum length ≥ 3 cm, repeat OGD every 2–3 years.	Repeat endoscopy in 6 months. If LGD is confirmed, endoscopic ablation should be offered. If ablation is not undertaken, 6-monthly surveillance.	OGD in tertial referral centre. If macroscopically visible lesion, endoscopic resection and RFA If flat lining, RFA treatment.
	ESGE	Weusten et al., 2017 [48]	If columnar-lined oesophagus < 1 cm, no surveillance. If BO ≥1 cm and < 3 cm, repeat OGD every 5 years. If BO ≥ 3 cm and < 10 cm, repeat OGD every 3 years. If BO ≥10 cm, refer to a BO expert centre.	Repeat OGD at a BO expert centre in 6 months. If no dysplasia is found, repeat after 1 year. After two subsequent endoscopies negative for dysplasia, follow standard surveillance for ND-BO. If LGD is confirmed, endoscopic ablation should be offered.	All visible abnormalities should be removed by endoscopic resection techniques. If no suspicious visible lesions, take biopsies; if negative for dysplasia, repeat endoscopy at 3 months; if HGD confirmed, endoscopic ablation, preferably with RFA.
American	AGA	Sharma et al., 2020 [49]	No endoscopic treatment indicated. No indications on endoscopic surveillance.	Repeat examination within 3–6 months to rule out visible lesions, which should prompt endoscopic resection. Both endoscopic therapy and continued surveillance are reasonable options for the management of LGD-BO.	Flat HGD should prompt a repeat HD-WLE (6–8 weeks) to evaluate for the presence of a visible lesion; these visible lesions should be removed by EMR. Endoscopic therapy is the preferred treatment over oesophagectomy.
	ACG	Shaheen et al., 2022 [50]	If < 1 cm salmon-coloured mucosa or irregular Z-line, no biopsy. If BO < 3 cm length, repeat OGD every 5 years. If BO ≥ 3 cm length, repeat OGD every 3 years.	Discuss risks and benefits of surveillance vs endoscopic therapy. If surveillance, endoscopy every 6 months for one year, then annually. If endoscopic therapy, resection of all visible lesions followed by ablation of the remaining BO.	Endoscopic resection of all visible lesions followed by ablation of the remaining BO.
Asian Pacific	Asia-Pacific consensus	Fock et al., 2016 [51]	No proven benefit in endoscopic surveillance of BO in the absence of dysplasia. If surveillance, OGD every 3–5 years with biopsy protocol.	Consider treatment or surveillance. If treatment, resect visible lesions. In the absence of focal lesions, consider RFA. If surveillance, repeat endoscopy in 6 months to confirm LGD.	Endoscopic resection for BO with HGD and carcinoma in situ when visible lesions. RFA to ablate all BO. Surgery can be an alternative to endoscopic resection (with or without RFA).

BSG: British Society of Gastroenterology, ESGE: European Society of Gastrointestinal Endoscopy, AGA: American Gastroenterological Association, ACG: American College of Gastroenterology, ND-BO: non-dysplastic Barrett’s oesophagus, LGD-BO: low-grade Barrett’s oesophagus, HGD-BO: high-grade Barrett’s oesophagus, OGD: oesophagogastroduodenoscopy, RFA: radiofrequency ablation

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
