# Peer review of "Endoscopic Management of Dysplastic Barrett’s Oesophagus and Early Oesophageal Adenocarcinoma"

_cancers, 2023, doi:10.3390/cancers15194776_

Round 1
Reviewer 1 Report
The text provides a concise and informative summary of the topic "Endoscopic Resection Techniques for Dysplastic Barrett’s and Early Oesophageal Adenocarcinoma." It effectively highlights key points such as the importance of endoscopic resection (ER) in managing Barrett's esophagus and early adenocarcinoma, the differences between EMR and ESD techniques, and the significance of combining ER with ablation to prevent recurrence.
The text is well-structured, making it easy for readers to grasp the essential concepts without unnecessary complexity. It also mentions common complications associated with ER, offering a balanced perspective on the procedure's challenges.
In conclusion, this text should be accepted as it provides a clear and concise overview of the subject matter.
Author Response
The text provides a concise and informative summary of the topic "Endoscopic Resection Techniques for Dysplastic Barrett’s and Early Oesophageal Adenocarcinoma." It effectively highlights key points such as the importance of endoscopic resection (ER) in managing Barrett's esophagus and early adenocarcinoma, the differences between EMR and ESD techniques, and the significance of combining ER with ablation to prevent recurrence.
The text is well-structured, making it easy for readers to grasp the essential concepts without unnecessary complexity. It also mentions common complications associated with ER, offering a balanced perspective on the procedure's challenges.
In conclusion, this text should be accepted as it provides a clear and concise overview of the subject matter.
Reply: We thank the reviewer for the nice comments. No revisions needed.
Reviewer 2 Report
Nice review regarding the Barrett Esophagus. In my opinion, I would try to add more details regarding the virtual chromoendoscopy adding more detailed characteristics and description of BO. I would do the same for endomicroscopy. Maybe you should consider adding some suggestive images for that.
Author Response
Nice review regarding the Barrett Esophagus. In my opinion, I would try to add more details regarding the virtual chromoendoscopy adding more detailed characteristics and description of BO. I would do the same for endomicroscopy.
Reply: we thank the reviewer for his comments. As suggested, we have expanded the paragraph on virtual chromoendoscopy and on endomicroscopy in the "Diagnosis of Barrett's oesophagus" section.
Maybe you should consider adding some suggestive images for that.
Reply: we have now added an image on virtual chromoendoscopy as suggested (Figure 1). Unfortunately, we do not have any endomicroscopy images available at the moment.
Reviewer 3 Report
Table 1 is not illustrative enough!!! It is better if you can make 3 algorithms, one for Europe, America, and Asia.
Line 256: difference in in neoplastic-----> Only one in.
It would help if you documented your preference and what is going on in each case; LGD, HGD whether short or long segments, and OAC in your centers.
You should add a section or paragraph for medical management after the endoscopic procedure.
Author Response
Table 1 is not illustrative enough!!! It is better if you can make 3 algorithms, one for Europe, America, and Asia.
Reply: The purpose of the table was to highlight and compare the differences between the international guidelines for the management of Barrett’s oesophagus. Considering these differences, we believe that different guidelines cannot be combined in a single algorithm for each continent.
We agree with the reviewer that an algorithm might be visually appealing, therefore we have added an algorithm based on the BSG guidelines (Figure 2), as these are the ones that we currently follow in our clinical practice.
Line 256: difference in in neoplastic-----> Only one in.
Reply: We thank the reviewer for noticing the mistake. We have corrected the sentence accordingly.
It would help if you documented your preference and what is going on in each case; LGD, HGD whether short or long segments, and OAC in your centers.
Reply: We thank the reviewer for his comment. This review aimed to summarise the current evidence and guidelines indications as summarised in Table 1. In our centres we tend to follow the ESGE/BSG guidelines regarding BO follow-up and treatment for straight forward cases. However, we acknowledge that more complex cases require a case-by-case evaluation taking into account several factors including institutional experience, lesion’s characteristics and patient’s comorbidities, therefore we usually discussed such cases at the MDT meetings in order to decide the most appropriate management. We have added a paragraph at the end of the ablation section reporting this concept.
You should add a section or paragraph for medical management after the endoscopic procedure.
Reply: we agree with the reviewer that the medical management after endoscopic procedures is an important issue, although there are no clear indications are given by the current guidelines and the literature on this topic is not conclusive.
We have added a section on this topic at the end of the manuscript as suggested.